# Multi-agent Reinforcement Learning with Hybrid Action Space for Free Gait Motion Planning of Hexapod Robots

**Huiqiao Fu[1], Kaiqiang Tang[1,†], Peng Li[2], Guizhou Deng[1], Chunlin Chen[1,†]**
[1]Department of Control Science and Intelligent Engineering, Nanjing University
[2]Institute of Software, Chinese Academy of Sciences
[†]Corresponding author
{hqfu, kqtang, gzdeng}@smail.nju.edu.cn, lipeng@iscas.ac.cn, clchen@nju.edu.cn

**Abstract:** Legged robots are able to overcome challenging terrains through diverse gaits formed by contact sequences. However, environments characterized by discrete footholds present significant challenges. In this paper, we tackle the problem of free gait motion planning for hexapod robots walking in randomly generated plum blossom pile environments. Specifically, we first address the complexity of multi-leg coordination in discrete environments by treating each leg of the hexapod robot as an individual agent. Then, we propose the Hybrid action space Multi-Agent Soft Actor Critic (Hybrid-MASAC) algorithm capable of handling both discrete and continuous actions. Finally, we present an integrated free gait motion planning method based on Hybrid-MASAC, streamlining gait, Center of Mass (COM), and foothold sequences planning into a single model. Comparative and ablation experiments in both of the simulated and real plum blossom pile environments demonstrate the feasibility and efficiency of our method.

**Keywords:** Free Gait, Hexapod Robot, Hybrid Action Space, Multi-agent Reinforcement Learning

## 1 Introduction

Legged robots can navigate challenging environments using a variety of gaits defined by contact sequences [1, 2, 3]. Compared to bipedal and quadrupedal robots, hexapod robots possess advantages such as more flexible gaits and higher stability, making them applicable to a variety of fields, such as disaster rescue, material transportation, planet exploration, and more [4, 5, 6]. However, in extreme conditions like discrete environments, these robots encounter a major challenge due to the limited foothold choices, and need meticulous planning for selecting reliable gait, Center of Mass (COM), and foothold sequences. In this paper, we address the challenge of automatically generating reliable gait, COM, and foothold sequences for hexapod robots in uneven plum blossom pile environments with a single model.

During locomotion, each leg of the hexapod robot undergoes alternating support and swing phases, generating diverse gait patterns through phase combinations. These gaits are categorized into periodic and aperiodic types based on their movement rhythm [7]. Gait planning for hexapod robots often involves formulating sequences based on rules or heuristics [8]. Additionally, biologically inspired methods like the Central Pattern Generator (CPG) can control gait rhythmically without sensory or descending inputs [9, 10]. However, these approaches typically demand significant manual design effort. In COM trajectory planning, methods usually fall into two categories: discrete search and continuous optimization. Discrete search methods involve a successor set and a list of possible poses for one state relative to another, forming a tree of trajectories explored to find a path from start to goal [11, 12]. While, in high-dimensional spaces, these methods encounter significant

8th Conference on Robot Learning (CoRL 2024), Munich, Germany.

challenges, termed the "curse of dimensionality". On the contrary, continuous optimization aims to find optimal motion trajectories by solving a set of optimization objectives and constraints, typically assuming convexity [13, 14]. However, it is difficult to establish an accurate mathematical model of the problem or find a feasible solution based on these methods. [15].

In recent years, Deep Reinforcement Learning (DRL) algorithms, along with Multi-Agent Reinforcement Learning (MARL), have shown great potential for intelligent decision-making in high-dimensional state-action spaces, and have been widely used in the field of robotics [16, 17, 18, 19, 20, 21]. For instance, Tsounis et al. [22] combined DRL with model-based motion planning, formulating the MDP by evaluating dynamic feasibility criteria instead of relying on physical simulation for motion planning in challenging terrains. Fu et al. [23] achieved multi-contact motion planning for a hexapod robot in uneven plum blossom pile environments, demonstrating feasibility in both simulation and reality. However, these methods, relying on fixed gait patterns, may result in poor passibility in complex environments. Recently, some researchers addressed the free gait motion planning problem for hexapod robots in the plum blossom pile environment using a hierarchical framework, utilizing the DRL method to generate the COM sequence and pre-defined rules to select the foothold sequence [24, 25]. Yet, most existing methods plan gait, COM, and foothold sequences independently, presenting two primary challenges in planning the overall motion of hexapod robots in complex environments. Firstly, decision variables encompass both discrete actions (e.g., leg phases) and continuous actions (e.g., COM position, foothold), making most RL algorithms unsuitable. Secondly, the numerous combinations of foothold points in the environment lead to an exponential growth of the action space with the number of robot legs.

In this paper, we address the challenge of free gait motion planning for hexapod robots in uneven plum blossom pile environments using a unified model. To simplify the complex action space, each leg is treated as an independent agent, resulting in an action space independent of the number of legs. These agents aim to collaborate to guide the robot's COM to a designated target area. We model the motion of all legs as a Markov game with a state transition function determined by the free gait Transition Feasibility Model (free gait TFM). Based on the Soft Actor Critic (SAC) algorithm [26], we propose the Hybrid action space Multi-Agent Soft Actor Critic (Hybrid-MASAC) algorithm, capable of handling both discrete and continuous actions in the free gait motion planning process. Finally, we present an integrated free gait motion planning method based on Hybrid-MASAC for hexapod robots moving in uneven plum blossom pile environments. Experimental results demonstrate that our method outperforms state-of-the-art fixed gait and free gait motion planning algorithms. Videos are shown at `http://www.hexapod.cn/marlhexa.html`. The contributions of this paper are threefold:

1. We formulate the free gait motion planning of the hexapod robot as a Markov game, and solve the optimization problem to generate optimal gait, COM, and foothold sequences.

2. We propose the Hybrid-MASAC algorithm to solve the multi-contact motion planning problem in the hybrid action space and the multi-agent framework.

3. We test the trained policies in different settings of plum blossom pile environments and compare the proposed Hybrid-MASAC-based motion planning method with the baseline methods. Both simulation and real-world experimental results demonstrate the feasibility and efficiency of the proposed method.

## 2 Preliminaries and Problem Description

### 2.1 Markov Game in Hybrid Action Space

The motion of the hexapod robot in the plum blossom pile environment can be formulated as a Markov game of a 6-tuple $\langle N, \mathbf{S}, \mathbf{A}, \mathbf{T}, \mathbf{R}, \mathbf{O} \rangle$, where $\mathbf{S}$ represents the state space containing the local observation of $N$ agents $\{\mathbf{O}_1, \cdots, \mathbf{O}_N\}$ and other environmental information. $\mathbf{A} = \{\mathbf{A}_1, \cdots, \mathbf{A}_N\}$ is the action space of $N$ agents. The state transition function $\mathbf{T} : \mathbf{S} \times \mathbf{A}_1 \times \ldots \times \mathbf{A}_N \to P(\mathbf{S})$ defines the distribution of the next state given the current state and the ac-

tion. $\mathbf{R} : \mathbf{S} \times \mathbf{A}_1 \times \ldots \times \mathbf{A}_N \to \mathbb{R}$ is the reward function. Each agent needs to learn a policy $\pi_i : \mathbf{O}_i \to P(\mathbf{A}_i)$ to map the local observation $\mathbf{O}_i$ to the action distribution $P(\mathbf{A}_i)$. The goal is to learn a policy $\pi_i$ that maximize the cumulative discounted returns $\mathbf{G}_{it} = \sum_{k=0}^{T-t} \gamma^k \mathbf{R}_{it+k}$, where $\gamma \in [0, 1]$ is the discount factor and $T$ is the time horizon.

When both the discrete action $\mathbf{a}^d$ and the continuous action $\mathbf{a}^c$ are included in the action space, the hybrid action space of the agent $i$ is defined as $\mathbf{A}_i^H = \{\mathbf{a}_i^d, \mathbf{a}_i^c\}$, where $\mathbf{a}_i^d = \{\mathbf{a}_{i1}^d, \cdots, \mathbf{a}_{iD}^d\}$ consists of $D$ discrete components and $\mathbf{a}_i^c = \{\mathbf{a}_{i1}^c, \cdots, \mathbf{a}_{iC}^c\}$ consists of $C$ continuous components. We assume that actions of the same type (discrete or continuous) are independent of each other, and the hybrid policy of the agent $i$ is decomposed as

$$\pi(\mathbf{a}_i^d, \mathbf{a}_i^c \mid \mathbf{o}_i) = \pi^d(\mathbf{a}_i^d \mid \mathbf{o}_i)\pi^c(\mathbf{a}_i^c \mid \mathbf{o}_i, \mathbf{a}^d) = \prod_{j=1}^{D} \pi^d(\mathbf{a}_{ij}^d \mid \mathbf{o}_i) \prod_{j=1}^{C} \pi^c(\mathbf{a}_{ij}^c \mid \mathbf{o}_i, \mathbf{a}^d). \quad (1)$$

In this paper, we represent each component of the discrete policy as a categorical distribution of $K$ possible categories and represent each component of the continuous policy as a $M$-dimensional diagonal Gaussian distribution.

## 2.2 Problem Description

The robot consists of an unactuated floating base and 6 legs, each with 3 actuated rotational joints, and the goal is to move from a random initial point $\mathbf{P}^{\text{initial}}$ to a random target area $\mathbf{P}^{\text{target}}$ with the shortest possible trajectory, while fulfilling all the kinematic and dynamic constraints. At each time step $t$, the state of the hexapod robot is defined as $\mathbf{s}_t := \langle \boldsymbol{\psi}_t, \mathbf{p}_t^r \rangle$, where $\boldsymbol{\psi}_t := \langle \mathbf{r}_{Bt}, \boldsymbol{\theta}_{Bt}, \mathbf{p}_t^c, \mathbf{c}_t \rangle$ is the proprioceptive information of the hexapod robot, $\mathbf{r}_{Bt} \in \mathbb{R}^3$ is the COM coordinates of the body, $\boldsymbol{\theta}_{Bt} \in \mathbb{R}^2$ represents the roll and pitch Euler angles of the body, $\mathbf{p}_t^c := \langle \mathbf{p}_{1t}^c, \cdots, \mathbf{p}_{6t}^c \rangle$ represents the foot position of the legs at time step $t$, $\mathbf{c}_t$ is the binary historical phase of each leg in the last three timesteps, where 0 indicates the swing phase and 1 indicates the support phase, and $\mathbf{p}_t^r = \mathbf{p}_t^c - \mathbf{p}_t^g$ represents the relative position of the current foot position and the target foot position.

The hybrid action $\mathbf{a}_t$ consists of the discrete action $\mathbf{a}_{it}^d$ and the continuous action $\mathbf{a}_{it}^c := \langle \mathbf{a}_{it}^{cs}, \mathbf{a}_{it}^{cc} \rangle$ of the leg $i, i \in \{1, \cdots, 6\}$. When $\mathbf{a}_{it}^d = 0$, the $i$-th leg is in swing phase, and the corresponding continuous action $\mathbf{a}_{it}^{cs}$ determines the next foothold position. When $\mathbf{a}_{it}^d = 1$, the $i$-th leg is in support phase, and the corresponding continuous action $\mathbf{a}_{it}^{cc}$ determines the next position and the next Euler angles of the body. Distinct phase combinations give rise to various gait types. The objective is to find an optimal policy that outputs hybrid actions for the hexapod robot, so as to generate optimal gait, COM, and foothold sequences.

To ensure the traceability of the output trajectory sequences, we expand upon the fixed gait Transition Feasibility Model (fixed gait TFM) proposed by Fu et al. [23] to create a free gait version. This free gait TFM, denoted as $T(\mathbf{s}_t, \mathbf{a}_t, \mathbf{s}_{t+1}')$, evaluates whether two adjacent states can be transitioned. $T(\mathbf{s}_t, \mathbf{a}_t, \mathbf{s}_{t+1}') = 1$ indicates that the state transition is feasible, and the robot successfully progresses to the next state $\mathbf{s}_{t+1}$. Conversely, $T(\mathbf{s}_t, \mathbf{a}_t, \mathbf{s}_{t+1}') = 0$ indicates that the state transition is unfeasible, and the robot returns to the last state $\mathbf{s}_t$ (please refer to the Appendix C for further details). The agents collectively aim to guide the robot's COM to a designated target area within the Markov game framework, while satisfying all kinematic and dynamic constraints.

# 3 Methodology

## 3.1 Overall Control Structure

The hexapod robot aims to navigate from an initial point to a target area in the uneven plum blossom pile environment, utilizing a free gait while minimizing the traveled distance. Each leg of the hexapod robot undergoes alternating support and swing phases, where swing phase legs select the next foothold position, while support phase legs adjust the body's direction and step size. This iterative process generates sequences for gait, COM, and footholds. The overall control structure is illustrated

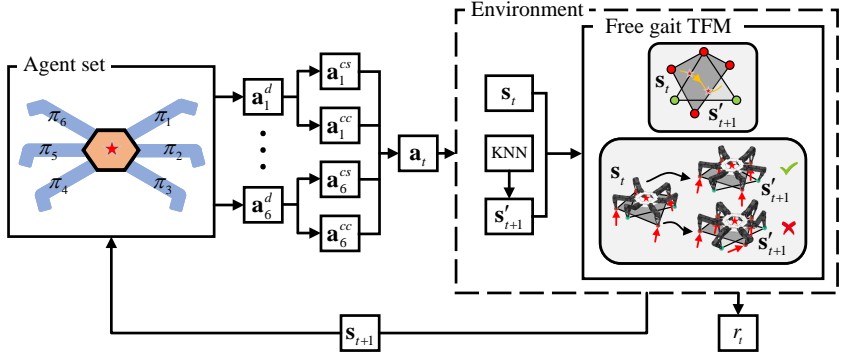

Figure 1: Overview of the proposed control structure. Each leg of the hexapod robot is regarded as an independent agent with a policy symbolized as $\pi_i, i \in \{1, \cdots, 6\}$.

in Fig. 1. Each agent, denoted as $\pi_i, i \in \{1, \cdots, 6\}$, receives local observations $\mathbf{o}_i$ and generates corresponding actions $\mathbf{a}_i$. To determine the phase of each agent $i$ within a step cycle, $\pi_i$ initially produces the discrete action $\mathbf{a}_i^d$, parameterized as a categorical distribution with two possible categories. If $\mathbf{a}_i^d = 0$, agent $i$ enters the swing phase, and $\pi_i$ generates the corresponding continuous action $\mathbf{a}_i^{cs}$, governing the selection of the next foothold. Conversely, if $\mathbf{a}_i^d = 1$, agent $i$ enters the support phase, and $\pi_i$ produces the relevant continuous action $\mathbf{a}_i^{cc}$, controlling the movement of the COM. These discrete and continuous actions collectively form the hybrid action $\mathbf{a}_t$. Based on $\mathbf{a}_t$, the environment calculates the position of each foot in the swing phase, then employs the K-Nearest Neighbor (KNN) algorithm to identify the nearest plum blossom pile as the target foothold. Subsequently, the next COM position and Euler angles are determined based on the actions generated by all support phases. The resulting next pending state $\mathbf{s}'_{t+1}$ for the hexapod robot is then generated.

Given the current state $\mathbf{s}_t$ and the next pending state $\mathbf{s}'_{t+1}$, a free gait Transition Feasibility Model (free gait TFM), denoted as $T(\mathbf{s}_t, \mathbf{a}_t, \mathbf{s}'_{t+1})$, assesses the transition feasibility between the two states. More details about the free gait TFM can be found in Appendix C. A feasible transition allows the hexapod robot to advance to the next state, $\mathbf{s}_{t+1} = \mathbf{s}'_{t+1}$, while an unfeasible transition prompts the robot to revert to the previous state, $\mathbf{s}_{t+1} = \mathbf{s}_t$. Finally, the environment outputs the next state $\mathbf{s}_{t+1}$ and computes the reward $r_t$. When reaching the target area or surpassing the maximum number of steps, the current episode is terminated, and the hexapod robot is reset to a new initial point, with a new target area generated to start a new episode. The optimal policy $\pi_i^*$ of each agent is trained by the proposed Hybrid-MASAC algorithm, detailed in Subsection 3.2, and the optimal gait, COM, and foothold sequences of the hexapod robot are finally output by the optimal policy set $\pi^* := \langle \pi_1^*, \cdots, \pi_6^* \rangle$.

### 3.2 Hybrid-MASAC for Free Gait Motion Planning

In multi-agent environment, each agent updates its policy independently during the learning process, resulting in a non-stationary environment that violates the MDP hypothesis conditions. Thus, we use the framework of Centralized Training with Decentralized Execution (CTDE) [27] to solve this problem. Furthermore, to address the decision making problem in the hybrid action space that includes both discrete and continuous actions, we draw inspiration from the work of Delalleau et al.[28], and further perform the hybrid policy iteration within the multi-agent framework, yielding the Hybrid action space Multi-agent Soft Actor Critic (Hybrid-MASAC) algorithm. The network architecture is illustrated in Figure 2, and the pseudocode of the integrated Hybrid-MASAC-based free gait motion planning method is detailed in *Algorithm* 1.

In the Hybrid-MASAC algorithm, the hybrid policy is defined as

$$\pi(\mathbf{a} \mid \mathbf{s}) = \pi(\mathbf{a}^c, \mathbf{a}^d \mid \mathbf{s}) = \pi^c(\mathbf{a}^c \mid \mathbf{s}, \mathbf{a}^d)\pi^d(\mathbf{a}^d \mid \mathbf{s}), \tag{2}$$

where $\mathbf{a}^d$ represents the discrete actions, $\mathbf{a}^c$ represents the continuous actions, $\pi^d$ is the discrete policy distribution, and $\pi^c$ is the continuous policy distribution. The joint entropy of the hybrid policy is calculated as

$$\pi^d \left( \cdot \mid \mathbf{s} \right)^T \mathcal{H} \left( \pi^c \left( \cdot \mid \mathbf{s}, \mathbf{a}^d \right) \right) + \mathcal{H} \left( \pi^d \left( \cdot \mid \mathbf{s} \right) \right). \quad (3)$$

Based on the hybrid policy, the joint entropy, and the multi-agent framework, we redefine the soft state-value function of the SAC for the i-th agent as

$$
\begin{aligned}
V_{\boldsymbol{\theta}_i} \left( \mathbf{o}_t \right) = \mathbb{E}_{\mathbf{a}_t^c \sim \pi^c} & \left[ \pi_{\boldsymbol{\phi}_i}^d \left( \cdot \mid \mathbf{o}_{it} \right)^T \left( Q_{\boldsymbol{\theta}_i} \left( \mathbf{o}_t, \cdot, \mathbf{a}_t^c \right) \right. \right. \\
& - \alpha_i^d \log \pi_{\boldsymbol{\phi}_i}^d \left( \cdot \mid \mathbf{o}_{it} \right) \\
& \left. \left. - \alpha_i^c \log \pi_{\boldsymbol{\phi}_i}^c \left( \mathbf{a}_{it}^c \mid \mathbf{o}_{it}, \mathbf{a}_{it}^d \right) \right) \right],
\end{aligned}
$$
$$(4)$$

where $\pi_{\phi^d}$ is the discrete policy, $\pi_{\phi_i^c}$ is the continuous policy, $\boldsymbol{\theta}_i$ and $\boldsymbol{\phi}_i$ are Q-network parameters and policy network parameters respectively, $\mathbf{o} = \{\mathbf{o}_1, \cdots, \mathbf{o}_N\}$ is the local observation set, $\mathbf{a}^d = \{\mathbf{a}_1^d, \cdots, \mathbf{a}_N^d\}$ is the discrete action set, $\mathbf{a}^c = \{\mathbf{a}_1^c, \cdots, \mathbf{a}_N^c\}$ is the continuous action set, $\alpha_i^d$ is the temperature for the discrete policy, and $\alpha_i^c$ is the temperature for the continuous policy.

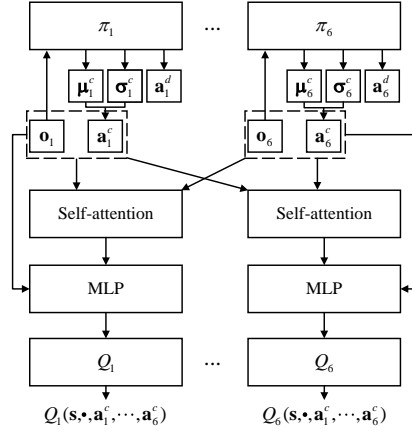

Figure 2: Hybrid-MASAC architecture. The policy $\pi_i$ initially outputs the discrete action $\mathbf{a}_i^d$, and the parameters of the continuous action distribution $\boldsymbol{\mu}_i^c, \boldsymbol{\sigma}_i^c$, based on the input observation $\mathbf{o}_i$. Next, all the observation and continuous action pairs are fed into the self-attention mechanism, and the resulting output features are added to the observation and continuous action pairs as the input of the MLP. Finally, the Q-value $Q_i \left( \mathbf{s}, \cdot, \mathbf{a}_1^c, \cdots, \mathbf{a}_6^c \right)$ is output by the MLP.

We incorporate an attention mechanism [29] to allocate attention within the Q-network to different agent behaviors. The Q-function, enhanced by this attention mechanism, is computed as follows:

$$Q_{\boldsymbol{\theta}_i} \left( \mathbf{o}, \cdot, \mathbf{a}^c \right) = f_i \left( g_i \left( \mathbf{o}_i, \cdot, \mathbf{a}_i^c \right), \mathbf{v}_i \right), \quad (5)$$

where $g_i$ is a single-layer fully connected network, which outputs state-action feature vectors according to the state-action pair of the agent $i$, $f_i$ is a two-layer fully connected network that outputs Q-values according to the feature vectors concatenating by $v_i$ and $g_i \left( \mathbf{o}_i, \cdot, \mathbf{a}_i^c \right)$, $v_i$ is the weighted sum of the state-action feature vectors of all agents except agent $i$, which is calculated as

$$\mathbf{v}_i = \sum_{j \neq i} \alpha_{i,j} g_i \left( \mathbf{o}_j, \cdot, \mathbf{a}_j^c \right), \quad (6)$$

The attention weight $\alpha_{i,j}$ is calculated as

$$\alpha_{i,j} = \frac{\exp \left( \beta_{i,j} \right)}{\sum_{j \neq i} \exp \left( \beta_{i,j} \right)}, \beta_{i,j} = g_i^T \left( \mathbf{o}_i, \cdot, \mathbf{a}_i^c \right) W_k^T W_q g_i \left( \mathbf{o}_j, \cdot, \mathbf{a}_j^c \right), \quad (7)$$

where $\beta_{i,j}$ represents the correlation between the state-action feature vectors of agent $i$ and other agents, $W_k$ and $W_q$ are the weight matrices to be learned, $\alpha_{i,j}$ is the result of softmax on $\beta_{i,j}$.

The policy evaluation of the Hybrid-MASAC is performed by minimizing the redefined Behrman residuals:

$$J_Q(\boldsymbol{\theta}_i) = \mathbb{E}_{\left( \mathbf{o}_t, \mathbf{a}_t^d, \mathbf{a}_t^c \right) \sim \mathcal{D}} \left[ \frac{1}{2} \left( Q_{\boldsymbol{\theta}_i} \left( \mathbf{o}_t, \mathbf{a}_t^d, \mathbf{a}_t^c \right) - \left( r_{it} + \gamma \mathbb{E}_{\mathbf{o}_{t+1} \sim p} \left[ V_{\bar{\boldsymbol{\theta}}_i} \left( \mathbf{o}_{t+1} \right) \right] \right) \right)^2 \right], \quad (8)$$

where $V_{\bar{\boldsymbol{\theta}}_i} \left( \mathbf{o}_{t+1} \right)$ is calculated by the target value network parameters $\bar{\boldsymbol{\theta}}_i$ and target policy network parameters $\bar{\boldsymbol{\phi}}_i$. The target function in the policy improvement of the Hybrid-MASAC is redefined as

$$
\begin{aligned}
J_\pi(\boldsymbol{\phi}_i) = \mathbb{E}_{\mathbf{o}_t \sim \mathcal{D}} & \left[ \mathbb{E}_{\mathbf{a}_t^c \sim \pi^c} \left[ \pi_{\boldsymbol{\phi}_i}^d (\cdot \mid \mathbf{o}_{it})^T \left( \alpha_i^d \log \pi_{\boldsymbol{\phi}_i}^d \left( \cdot \mid \mathbf{o}_{it} \right) \right. \right. \right. \\
& \left. \left. \left. + \alpha_i^c \log \pi_{\boldsymbol{\phi}_i}^c \left( \mathbf{a}_{it}^c \mid \mathbf{o}_{it}, \mathbf{a}_{it}^d \right) - Q_{\boldsymbol{\theta}_i} \left( \mathbf{o}_t, \cdot, \mathbf{a}_t^c \right) \right) \right] \right].
\end{aligned}
$$
$$(9)$$

The choice of the temperatures $\alpha^d$ and $\alpha^c$ directly impact the resulting policy. To enable an automatic adjustment of these temperatures, we draw inspiration from the temperature update method introduced by Haarnoja et al. [26] and extend it to accommodate the multi-agent framework and hybrid action space. For agent $i$, the temperature target functions for the continuous and discrete policies are defined as follows:

$$J(\alpha_i^d) = \pi_{\boldsymbol{\phi}_i}^d(\cdot \mid \mathbf{o}_{it})^T(-\alpha_i^d(\log \pi_{\boldsymbol{\phi}_i}^d(\cdot \mid \mathbf{o}_{it}) + \bar{H}^d)), \tag{10}$$

$$J(\alpha_i^c) = \mathbb{E}_{\mathbf{a}_{it}^c \sim \pi_{\boldsymbol{\phi}_i}^c}[-\alpha_i^c(\log \pi_{\boldsymbol{\phi}_i}^c(\mathbf{a}_{it}^c \mid \mathbf{o}_{it}, \mathbf{a}_{it}^d) + \bar{H}^c)], \tag{11}$$

where $\bar{H}^d$ and $\bar{H}^c$ represent two hyperparameters for discrete and continuous policy, respectively. In our experiments, we set $\bar{H}^d = 0.25$ and $\bar{H}^c = -0.25$.

## 4 Experiments

### 4.1 Training Setup

Three different types of simulated plum blossom pile environments, denoted as $E_1$, $E_2$, $E_3$, and a real plum blossom pile environment, $E_4$, are constructed to evaluate the proposed Hybrid-MASAC-based free gait motion planning method, as illustrated in Fig. 3. Each environment has dimensions of $1.2\ m$ in length and width. The number of plum blossom piles in each environment is denoted as $N_p$. For each simulated environment, we set $N_p$ to be $300, 200$, and $150$, respectively, resulting in a total of 9 different simulated environments.

We use 3 metrics to evaluate the performance of the proposed motion planning method: Average Episode Reward (AER), Average Episode Step (AES) and Average Success Rate (ASR). AER is calculated as the ratio of the total reward to the total number of steps in an episode. AES is determined as the ratio of the total steps to the distance between the initial point and the target area. ASR represents the average success rate observed in the last 200 episodes.

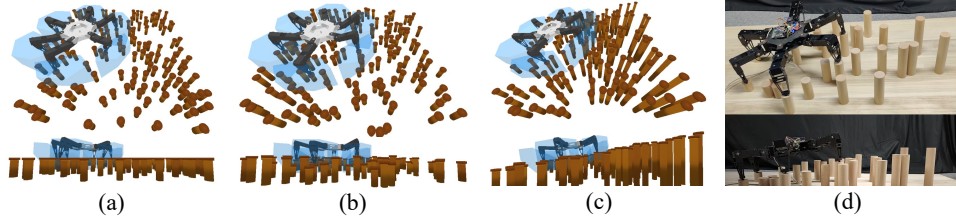

|      |      |      |      |
| :--: | :--: | :--: | :--: |
| (a)  | (b)  | (c)  | (d)  |

Figure 3: Different types of plum blossom pile environment with $N_p = 150$. (a) Randomly distributed plum blossom piles with the same height $E_1$. (b) Randomly distributed plum blossom piles with random height $E_2$. (c) Stair shape plum blossom pile environment $E_3$. (d) A simplified version of $E_3$ in the real-world $E_4$. The transparent blue polyhedrons in the simulated environments are the simplified foot workspace.

### 4.2 Experimental Results

We use the proposed Hybrid-MASAC algorithm to train the optimal hybrid policies in 9 different simulated environments. Fig. 4 shows the top view of the plum blossom pile environment used in the training process. Where, the green block is the initial point, the red circle is the target area, yellow dots represent plum blossom piles, red dots represent footholds and CoM sequences, and blue wireframes represent the body of the hexapod robot. As training progresses, all agents gradually learn to cooperate in guiding the robot to the target area, resulting in higher average rewards, fewer steps taken on average, and an increasing average success rate, as shown in Fig. 5. At the end of the training, the optimal policy can leads the hexapod robot to the target area with the shortest trajectory as shown in Fig. 4(b).

To evaluate the trained policies, we randomly set 100 groups of initial points and target areas in each environment, and compute the corresponding AER, AES and ASR metrics. The results are shown in Table 1. The optimal hybrid policies trained with Hybrid-MASAC can generate effective sequences of gait, COM, and foothold, and guide the hexapod robot to the target area with ASR exceeding 80%. When the number of plum blossom piles exceeds 200, the ASR approaches 100%. Conversely, as the number of plum blossom piles decreases, the ASR decreases due to the lack of suitable footholds for certain legs.

To comprehensively evaluate the performance of the proposed Hybrid-MASAC algorithm, we conduct a comparative analysis against several baseline meth-

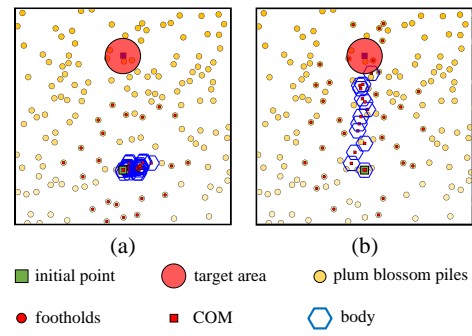

(a) (b)

initial point ◼  target area ●  plum blossom piles ○

footholds ●  COM ◼  body ⬡

Figure 4: Top view of the plum blossom pile environment used in the training process. (a) The beginning of the training. (b) The end of the training.

ods. These include 2 traditional graph-based methods: (a) A*-Tripod and (b) A*-Free, as well as 4 DRL-based methods: (c) Multi-agent Soft Actor Critic (MASAC) [30], a simplified counterpart to Hybrid-MASAC in the continuous action space, (d) Single-agent Soft Actor Critic with free gait (SAC-Free) [26], (e) Single-agent Soft Actor Critic with tripod gait (SAC-Tripod), and (f) HFG-DRL [24], a hierarchical DRL-based algorithm for free gait motion planning. More details about the above methods can be found in Appendix D.

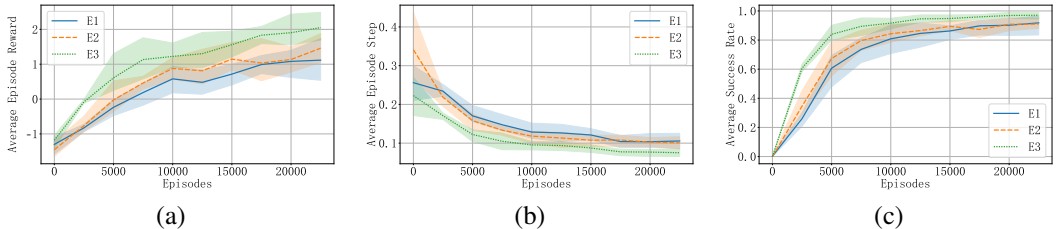

(a) (b) (c)

Figure 5: Learning curves of $E3$ with $N_p = 150$ (5 random seeds). (a) The learning curve of AER. (b) The learning curve of AES. (c) The learning curve of ASR.

Table 1: Metrics of the trained policies in 9 different types of environments using Hybrid-MASAC.

| $N_p$ | Metrics | $E_1$ | $E_2$ | $E_3$ |
|---|---|---|---|---|
| 300 | AER | $1.196_{\pm 0.460}$ | $2.575_{\pm 0.514}$ | $4.131_{\pm 0.469}$ |
| | AES | $0.071_{\pm 0.022}$ | $0.057_{\pm 0.012}$ | $0.042_{\pm 0.017}$ |
| | ASR | $98\%_{\pm 1\%}$ | $100\%_{\pm 0\%}$ | $100\%_{\pm 0\%}$ |
| 200 | AER | $1.798_{\pm 0.329}$ | $2.311_{\pm 0.513}$ | $2.628_{\pm 0.290}$ |
| | AES | $0.079_{\pm 0.025}$ | $0.063_{\pm 0.021}$ | $0.057_{\pm 0.016}$ |
| | ASR | $97\%_{\pm 2\%}$ | $99\%_{\pm 1\%}$ | $99\%_{\pm 1\%}$ |
| 150 | AER | $0.504_{\pm 0.358}$ | $1.805_{\pm 0.412}$ | $1.867_{\pm 0.422}$ |
| | AES | $0.133_{\pm 0.058}$ | $0.091_{\pm 0.021}$ | $0.085_{\pm 0.014}$ |
| | ASR | $83\%_{\pm 9\%}$ | $92\%_{\pm 5\%}$ | $96\%_{\pm 2\%}$ |

For a fair comparison with the motion planning method proposed by Fu et al. [23], we substitute the PPO algorithm with the SAC algorithm in the tripod gait case. The evaluation takes place within identical environmental conditions, maintaining consistent hyperparameters, such as learning rate, temperature, discount factor, et al. Table 2 presents a comparison of the metrics between baseline methods. For each experiments, we perform training with 5 different random seeds and use 100 groups of initial points and target areas for evaluation. The A*-Free algorithm benefits from free gait and achieves a higher success rate compared to A*-Tripod; however, due to the COM being

Table 2: Comparison of Hybrid MASAC and baseline methods in $E_3$ with $N_p = 150$.

| Algorithms | AER | AES | ASR |
|---|---|---|---|
| A*-Tripod | - | $0.257_{\pm 0.021}$ | $21\%_{\pm 5\%}$ |
| A*-Free | - | $0.189_{\pm 0.018}$ | $42\%_{\pm 3\%}$ |
| MASAC [30] | $-0.696_{\pm 0.223}$ | $0.215_{\pm 0.013}$ | $46\%_{\pm 5\%}$ |
| SAC-Free [26] | $-0.911_{\pm 0.180}$ | $0.231_{\pm 0.019}$ | $32\%_{\pm 9\%}$ |
| SAC-Tripod [23] | $1.652_{\pm 0.385}$ | $0.122_{\pm 0.018}$ | $90\%_{\pm 4\%}$ |
| HFG-DRL [24] | $0.355_{\pm 0.266}$ | $0.202_{\pm 0.015}$ | $58\%_{\pm 5\%}$ |
| Hybrid-MASAC (Ours) | $\mathbf{1.867}_{\pm 0.422}$ | $\mathbf{0.085}_{\pm 0.014}$ | $\mathbf{96\%}_{\pm 2\%}$ |

represented as discrete grids, the success rates of such algorithms are generally low. The HFG-DRL algorithm also has a relatively low success rate when the number of piles is only 150, as it selects gaits based on rules. The proposed Hybrid-MASAC algorithm outperforms all the baseline methods across all metrics, showcasing the feasibility and efficiency of our method in the planning of free gait motion for the hexapod robot walking in complex plum blossom pile environments.

Finally, we conducted the experiment in the real-world environment $E_4$. The video can be found in http://www.hexapod.cn/marlhexa.html. We assume access to environmental information and global COM coordinates. The trained hybrid policy enables the generation of optimal gait, COM, and foothold sequences for the hexapod robot. The target joint rotation angles for each time step are calculated by inverse kinematics. Subsequently, a PD controller guides each joint of the hexapod robot to follow the target trajectory. An inherent challenge stems from error accumulation during prolonged movements in the real environment. Nevertheless, in our experiment, the size of the environment is relatively small, where cumulative errors remain within acceptable thresholds. Worthy of note, the random plum blossom pile environments can characterize any non-structured environment to a certain extent. Our method holds promise for deployment in more challenging real-world environments, offering safer and more reliable motion reference trajectories.

## 5  Limitations

In the real world experiment, we assume the environmental information is known; however, for dynamic real-world scenarios, real-time environmental perception and localization are necessary. To tackle this issue, advanced techniques such as Simultaneous Localization and Mapping (SLAM) can be introduced for precise mapping and positioning, although this falls beyond the scope of our paper. It's important to highlight that the motion planning method outlined in this paper holds applicability to quadruped and biped robots as well. We can adapt the approach by adjusting the state space, action space, and transition feasibility model accordingly. However, advanced and robust control methodologies such as Model Predictive Control (MPC), Linear Quadratic Regulator (LQR), and various trajectory tracking techniques is needed to accurately track the output trajectory sequences.

## 6  Conclusion

A free gait motion planning method is proposed in this paper, where each leg of the hexapod robot is regarded as an independent agent, with the common objective of moving the COM from a initial point to a designated target area in the uneven plum blossom pile environment. The motion of all legs is modeled as a Markov game with a specific state transition function determined by the proposed free gait TFM. Based on the multi-agent framework and the hybrid action space, we propose the Hybrid-MASAC algorithm to train and generate the optimal hybrid policy for each agent. These trained policies enable the generation of optimal gait, COM, and foothold sequences for the hexapod robot, while fulfilling all the kinematic and dynamic constraints. All the experimental results in both simulation and real-world demonstrate the feasibility and efficiency of the proposed method. Our future work will focus on training more robust policies and combining the proposed method with localization and map-building methods.

**Acknowledgments**

This work was supported in part by the National Natural Science Foundation of China (Nos. 62073160 & 72394363), the Nanjing Science and Technology Plan under Grant 202309018, and the Nanjing University Integrated Research Platform of the Ministry of Education-Top Talents Program.

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

# A  Pseudocode of Hybrid-MASAC for Free Gait Motion Planning

The integrated Hybrid-MASAC based free gait multi-contact motion planning algorithm is shown as in *Algorithm* 1.

---

**Algorithm 1:** Hybrid-MASAC for Free Gait Motion Planning

---

1: Initialize network parameters $\phi$, $\theta$, replay buffer $\mathcal{D}$.
2: Initialize target network parameters $\bar{\phi} \leftarrow \phi$, $\bar{\theta} \leftarrow \theta$.
3: **for** episode $k = 0, 1, 2, ..., M$ **do**
4:     Randomly initialize $\mathbf{P}^{\text{initial}}$ and $\mathbf{P}^{\text{target}}$.
5:     Reset the hexapod robot and obtain the state $\mathbf{s}_0$ and observation set $\mathbf{o}_0$.
6:     **for** step $t = 0, 1, 2, ..., T$ **do**
7:         **for** agent $i = 1, \cdots, 6$ **do**
8:             Sample discrete actions $\mathbf{a}_{it}^d \sim \pi_{\phi_i}^d \left( \mathbf{a}_{it}^d | \mathbf{o}_{it} \right)$.
9:             Sample continuous actions $\mathbf{a}_{it}^c \sim \pi_{\phi_i}^c \left( \mathbf{a}_{it}^c | \mathbf{o}_{it}, \mathbf{a}_{it}^d \right)$.
10:        **end for**
11:        Perform the hybrid action $\mathbf{a}_t$ and obtain the
            pending state $\mathbf{s}_{t+1}'$.
12:        **if** transition feasibility $T(\mathbf{s}_t, \mathbf{a}_t, \mathbf{s}_{t+1}') = 1$ **then**
13:            $\mathbf{s}_{t+1} = \mathbf{s}_{t+1}'$, obtain $\mathbf{o}_{t+1}$ and $r_t$.
14:        **else**
15:            $\mathbf{s}_{t+1} = \mathbf{s}_t$, obtain $\mathbf{o}_{t+1}$ and $r_t$.
16:        **end if**
17:        Store experience $(\mathbf{o}_t, \mathbf{a}_t, r_t, \mathbf{o}_{t+1})$ in $\mathcal{D}_k$.
18:    **end for**
19:    **for** agent $i = 1, \cdots, 6$ **do**
20:        $\theta_i \leftarrow \theta_i - \lambda_Q \hat{\nabla}_{\theta_i} J_Q (\theta_i)$,
21:        $\phi_i \leftarrow \phi_i - \lambda_\pi \hat{\nabla}_{\phi_i} J_\pi(\phi_i)$,
22:        $\alpha_i^d \leftarrow \alpha_i^d - \lambda \hat{\nabla}_{\alpha_i^d} J(\alpha_i^d), \alpha_i^c \leftarrow \alpha_i^c - \lambda \hat{\nabla}_{\alpha_i^c} J(\alpha_i^c)$,
23:        $\bar{\theta}_i \leftarrow \tau \theta_i + (1 - \tau) \bar{\theta}_i, \bar{\phi}_i \leftarrow \tau \phi_i + (1 - \tau) \bar{\phi}_i$.
24:    **end for**
25: **end for**

---

# B  Markov Game Formulation

## B.1  Observation Space

The observation $\mathbf{o}$ of the hexapod robot is composed of 6 local observations $\{\mathbf{o}_1, \cdots, \mathbf{o}_6\}$, which is a subset of the state space. The local observation $\mathbf{o}_i \in s$ is defined as

$$\mathbf{o}_i := \langle \boldsymbol{\psi}, \mathbf{p}_i^r \rangle, \tag{12}$$

where $\boldsymbol{\psi} := \langle \mathbf{r}_B, \boldsymbol{\theta}_B, \mathbf{p}^c, \mathbf{c}_i \rangle$ represents the proprioceptive information of the hexapod robot. $\mathbf{p}_i^r = \mathbf{p}_i^c - \mathbf{p}_i^g$ is the relative position between the current position $\mathbf{p}_i^c$ of the i-th foot and its corresponding target position $\mathbf{p}_i^g$ in the world frame. $\mathbf{p}_i^g$ is generated according to the coordinate of the target area.

## B.2  Action Space

Under the multi-agent framework, each leg of the hexapod robot is defined as an independent agent. At each time step $t$, the agent $i$ needs to choose the current discrete phase and the corresponding continuous action that should be performed under that phase. The action $\mathbf{a}_i$ of agent $i$, generated by its corresponding policy $\pi_i$, is defined as

$$\mathbf{a}_i := \langle \mathbf{a}_i^d, \mathbf{a}_i^{cs}, \mathbf{a}_i^{cc} \rangle, \tag{13}$$

where the discrete action $\mathbf{a}_i^d \in \{0, 1\}$ represents the next phase of the i-th leg. When $\mathbf{a}_i^d = 0$, leg $i$ is in swing phase, otherwise leg $i$ is in support phase. $\mathbf{a}_i^{cs} := \Delta\mathbf{r}_{Fi}$ represents the coordinate change of the swing phase, and $\mathbf{a}_i^{cc} := \langle\Delta\mathbf{r}_{Bi}, \Delta\boldsymbol{\theta}_{Bi}\rangle$ is the continuous action of the support phase. The coordinate change of the COM is calculated as $\Delta\mathbf{r}_B = \frac{1}{N_c}\sum_{i=0}^{N_c}\Delta\mathbf{r}_{Bi}$, and the Euler angle change of the body is calculated as $\Delta\boldsymbol{\theta}_B = \frac{1}{N_c}\sum_{i=0}^{N_c}\Delta\boldsymbol{\theta}_{Bi}$, where $N_c$ is the number of the support phase, and the yaw angle is assumed to remain unchanged in the motion process.

## B.3 Reward Function

For all agents, the objective is to work together to move the COM from an initial point to a target area via the shortest trajectory, while fulfilling all the kinematic and dynamic constraints. To achieve this task, the reward $r_i$ is defined as

$$r_{i,t} = r_{w,t} + r_{k,t} + r_{f,t} + r_{s,t} + r_{hi,t} + r_{di,t} + r_{g,t}, \tag{14}$$

where $r_{w,t}$ penalizes the improper combination of support phases, including two cases: the number of total support phases $N_{c,t} \leq 2$, and there is no support phase in the adjacent three legs. $r_{k,t}$ penalizes the hexapod robot for any leg that does not meet the kinematics constraint in the next state. $r_{f,t}$ penalizes the unfeasible state transition of hexapod robot according to the free gait TFM. $r_{s,t}$ penalizes the hexapod robot for every step it takes. $r_{hi,t}$ penalizes the leg $i$ for more than three consecutive time steps in the same phase to prevent the hexapod robot from standing still. $r_{di,t}$ rewards the agent $i$ for moving towards the target area. $r_{g,t}$ rewards the hexapod robot for reaching the target area.

## C Free gait Transition Feasibility Model

Referring to the fixed gait Transition Feasibility Model (TFM) of hexapod robots [23], we propose a free gait TFM $T(\mathbf{s}_t, \mathbf{a}_t, \mathbf{s}'_{t+1})$ to determine whether the transition between two adjacent states can be completed under all the kinematic and dynamic constraints, which includes three parts:

- $T_c(\mathbf{p}_t^c)$: Check for illegal phase combinations. For example, the number of total support phases is less than 3, or there is no support phase in the three adjacent feet.
- $T_k(\boldsymbol{\psi}_t, \boldsymbol{\psi}'_{t+1})$: Check for kinematic constraints.
- $T_d(\boldsymbol{\psi}_t, \boldsymbol{\psi}'_{t+1})$: Check for dynamic constraints.

where $\mathbf{s}'_{t+1}$ represents the next pending state. We formulate the transition feasibility as a nonlinear constrained optimization problem using the direct multiple shooting method and solve it by an open-source tool for nonlinear optimization, CasADi[1]. In CasADi, it uses interior point method to find the solution, and when the problem has no solution or the number of optimization steps reaches the preset maximum, we consider the problem to be infeasible.

## C.1 Phase Combination Constraints

The hexapod robot can use three or more support phases to form a support polygon to maintain good static stability. Thus, for the current state $\mathbf{s}_t$, we set phase combination constraints to filter out illegal phase combinations, including the number of total support phases $N_c \leq 2$ and the support phase number for the three adjacent feet should not be zero.

## C.2 Kinematic Constraints

For the initial state $s_0$ and each pending state $\mathbf{s}'_{t+1}$, the kinematic constraints are set up to avoid the robot's behavior violating the physical constraints of its own structure. The dimension parameters

---

[1] https://web.casadi.org/

Table 3: Dimensional parameters and joint rotation ranges of the hexapod robot.

| Parameter | Body | Coxa | Femur | Tibia |
|-----------|------|------|-------|-------|
| Length/mm | 240 | 60 | 120 | 145 |
| Range/° | - | [-45, 45] | [-45, 45] | [-135, -45] |

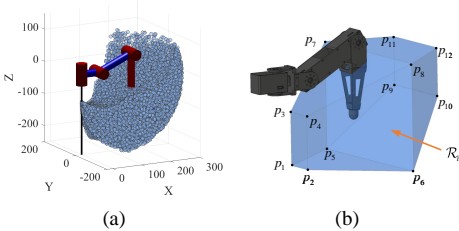

(a)                    (b)

Figure 6: The foot workspace of the hexapod robot. (a) The point cloud of original foot workspace. (b) Simplified foot workspace which is represented as a polyhedron.

and joint rotation ranges of the hexapod robot are shown in Table 3. By sampling within the joint rotation range, we can obtain the point cloud of the original foot workspace, as shown in Fig. 6(a). Since the original foot workspace is highly nonlinear and inconvenient for optimization, we conservatively simplify the kinematics constraint for leg $i$ as keeping its foot $\mathbf{p}_i^c$ within the polyhedron workspace $\mathcal{R}_i$, as shown in Fig. 6(b). For each plane in the polyhedron, we can find its normal vector $\mathbf{n}$ given the coordinates of three points. Here, we take $\mathbf{p}_1$, $\mathbf{p}_2$ and $\mathbf{p}_3$ in Fig. 6(b) as an example, the normal vector of this plane $\mathbf{n} = (\mathbf{p}_2 - \mathbf{p}_1) \times (\mathbf{p}_3 - \mathbf{p}_1)$. To keep the foot $\mathbf{p}_i^c$ inside the polyhedron, $\mathbf{p}_i^c$ needs to be on the right side of the plane, yielding the following constraint:

$$\mathbf{R}_Z\left(\alpha_i\right)\left[\left({}_W^B\mathbf{R}\mathbf{p}_i^c - \mathbf{p}_1\right) \cdot \mathbf{n}\right] > 0, \tag{15}$$

where ${}_W^B\mathbf{R}$ is the rotation matrix from the world frame to the body frame. Note that the polyhedron is described in the body frame and $\mathbf{p}_i^c$ is described in the world frame. $\mathbf{R}_Z(\alpha_i)$ is the z-axis rotation matrix and $\alpha_i$ is the z-axis deflection angle of the coxa frame to the body frame.

Each simplified foot workspace polyhedron consists of 12 planes. For each foot, there are a total of 12 kinematic constraints.

## C.3 Dynamic Constraints

In order to determine whether the hexapod robot can transfer from the current state $\mathbf{s}_t$ to the next pending state $\mathbf{s}'_{t+1}$, we set up the corresponding dynamic constraints and solve the problem as a mathematical feasibility problem using existing optimization tools. Specifically, we need to find a set of decision variables such as the COM position $\mathbf{r}_B \in \mathbb{R}^3$, the body Euler angles $\boldsymbol{\theta} \in \mathbb{R}^3$, the foot contact force $\mathbf{f} \in \mathbb{R}^{6\times3}$ et al., which fulfill all the constraints given $\mathbf{s}_t$ and $\mathbf{s}'_{t+1}$. The dynamic constraints include three parts: the dynamic model of the hexapod robot, the pushing force constraint and the friction cone constraint.

The dynamic model represents the time-dependent aspects of the system, and we approximate it by Single Rigid Body Dynamics (SRBD) [31]. Then, we get the Newton-Euler equations of the hexapod robot, which is defined as the SRBD:

$$m\ddot{\mathbf{r}}_B = \sum_{i=1}^N \mathbf{f}_i + m\mathbf{g}, \tag{16a}$$

$$\mathbf{I}\dot{\boldsymbol{\omega}} + \boldsymbol{\omega} \times \mathbf{I}\boldsymbol{\omega} = \sum_{i=1}^N \mathbf{f}_i \times \left(\mathbf{r}_B - \mathbf{p}_i^c\right), \tag{16b}$$

where $m$ is the mass of the hexapod robot, $\mathbf{f}_i \in \mathbb{R}^3$ is the contact force of foot $i$, $\mathbf{g} \in \mathbb{R}^3$ is the acceleration of gravity, $\boldsymbol{\omega} \in \mathbb{R}^3$ is the angular velocity of the body, $\mathbf{I} \in \mathbb{R}^{3\times3}$ is the inertia tensor

of the hexapod robot in the world frame, which can be calculated by the inertia tensor in the body frame $\mathbf{I}_B$ and the rotation matrix $_B^W\mathbf{R}$: $\mathbf{I} = {_B^W}\mathbf{R}\mathbf{I}_B{_B^W}\mathbf{R}^T$. $N$ is the number of the support foot. With free gait, $N = 6$ and we constrain the contact force of the swing phase to zero.

Using the SRBD, the state of the hexapod robot can be expressed by an ordinary differential equation $\dot{\mathbf{x}} = \mathbf{F}(\mathbf{x}, \mathbf{f})$, where $\mathbf{x} = [\mathbf{r}_B, \dot{\mathbf{r}}_B, \boldsymbol{\theta}_B, \boldsymbol{\omega}]^T$ is the state of the body which is only affected by the contact force. The rates of the Euler angles $\dot{\boldsymbol{\theta}}_B$ can be calculated by the optimized Euler angles $\boldsymbol{\theta}_B$ and the angular velocities $\dot{\boldsymbol{\omega}}$:

$$\dot{\boldsymbol{\theta}}_B = \mathbf{C}(\boldsymbol{\theta}_B)\dot{\boldsymbol{\omega}} = \begin{bmatrix} 1 & 0 & -\sin\theta_y \\ 0 & \cos\theta_x & \sin\theta_x\cos\theta_y \\ 0 & -\sin\theta_x & \cos\theta_x\cos\theta_y \end{bmatrix}\dot{\boldsymbol{\omega}}. \tag{17}$$

Using numerical integration methods, such as trapezoidal quadrature, we obtain the discrete dynamic constraint formula of the ordinary differential equation:

$$\mathbf{x}_{k+1} - \mathbf{x}_k \approx \frac{1}{2}(t_{k+1} - t_k)(\mathbf{F}(\mathbf{x}_{k+1}, \mathbf{f}_{k+1}) + \mathbf{F}(\mathbf{x}_k, \mathbf{f}_k)). \tag{18}$$

The pushing force constraints restrict the force provided by the environment to the hexapod robot can only be thrust:

$$\mathbf{f}_i \cdot \mathbf{n}\left(\mathbf{p}_i^c\right) \geq 0, \tag{19}$$

where $\mathbf{n}\left(\mathbf{p}_i^c\right)$ is the normal vector of the environmental surface at coordinate $\mathbf{p}_i^c$.

The friction cone constraint follows from Coulomb's law that pushing stronger into a surface allows exerting larger side-ways forces without slipping. Therefore, the resultant force on each support foot is always in the interior of the friction cone. The linear approximation is as follows:

$$\left|\mathbf{f}_i \cdot \mathbf{t}_{\{1,2\}}\left(\mathbf{p}_i^c\right)\right| \leq \boldsymbol{\mu} \cdot \mathbf{f}_i \cdot \mathbf{n}\left(\mathbf{p}_i^c\right), \tag{20}$$

where $\mathbf{t}_{\{1,2\}}\left(\mathbf{p}_i^c\right)$ is the tangential vector of the environment at coordinate $\mathbf{p}_i^c$ and $\boldsymbol{\mu}$ is the friction coefficient.

## D   Additional Experimental Details

At the beginning of a training episode, an initial point and a target area with a radius of $150\ mm$ are randomly generated, and the distance between them is greater than $300\ mm$. All agents collaborate to guide the hexapod robot's COM from the initial point to the target area, and the trained policies yield the optimal gait, COM, and foothold sequences.

The policy network is trained on a computer with an i7-7700 CPU and a Nvidia GTX 1060ti GPU. The networks are implemented using Pytorch[2], and the transition feasibility model used in the training process is solved using CasADi. All the simulated environments are built by PyBullet[3]. The hyperparameters of the Hybrid-MASAC algorithm are detailed in Table 4.

We conduct a comparative analysis against existing baseline methods: (a) A*-Tripod, an A* algorithm with the fixed tripod gait, using a 20x20 2-dim grid map to represent the robot's center of mass position in the plum blossom pile environment, and employing a fixed gait transition feasibility model to determine whether two nodes are transferable. (b) A*-Free, an A* algorithm with the free gait, utilizing a free gait transition feasibility model to assess the transferability of two nodes, with other settings identical to A*-Tripod. (c) Multi-agent Soft Actor Critic (MASAC) [30], a simplified counterpart to Hybrid-MASAC in the continuous action space, involves each agent $i$ generating a 10-dimensional continuous action vector, with each agent generating a 10-dimensional continuous action vector including 2 phase actions, 3 swing foot actions, 3 support foot actions, and 2 body Euler angle actions. (d) Single-agent Soft Actor Critic with free gait (SAC-Free) [26] utilizes a 35-dimensional continuous action vector for free gait motion planning, encompassing 6 groups

---
[2]https://pytorch.org/
[3]https://pybullet.org/

Table 4: Hybrid-MASAC hyperparameters during training.

| Parameter | Value |
|---|---|
| Optimizer | Adam |
| Learning rate | $10^{-2}$ |
| Discount factor | 0.95 |
| Replay buffer size | $10^{6}$ |
| Minibatch size | 1024 |
| Target smoothing coefficient | $10^{-2}$ |
| Maximum number of episode steps | 100 |
| Discrete policy target temperature | 0.25 |
| Continuous policy target temperature | -0.25 |

of 2-dimensional phase actions, 6 groups of 3-dimensional swing foot actions, alongside 3 COM actions and 2 body Euler angle actions. (e) Single-agent Soft Actor Critic with tripod gait (SAC-Tripod) uses a 14-dimensional continuous action vector for tripod gait, involving 3 COM actions, 2 body Euler angle actions, and 3 groups of 3-dimensional swing foot actions. (f) HFG-DRL [24]: A hierarchical DRL-based algorithm with the free gait, where the upper layer implements center of mass path planning and the lower layer implements motion planning, selecting gait types based on some rules.

# E   Additional Experimental Results

With the free gait generated by the policies, the hexapod robot can adapt to various gait patterns such as tripod gait, quadruped gait, and wave gait according to the surrounding environment. When the policy converges to the optimum, the proportion of the above three gaits in each environment is illustrated in Fig. 7. Notably, in the same environment, as the number of plum blossom piles decreases, to get a better passibility, the proportion of tripod gait decreases, while the proportion of wave gait increases. The proportion of the quadruped gait is relatively stable across all environments. Additionally, the distribution of gait types varies among different environments.

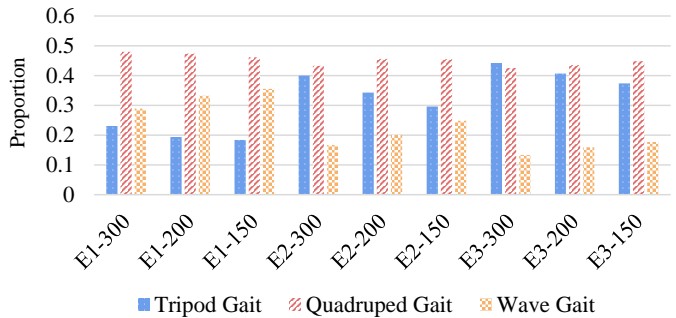

Figure 7: Statistics of gait types in 9 environments with free gait.

Fig. 8 shows the coxa angle curves of the hexapod robot obtained by the trained policy in environment $E_3$ with $N_p = 150$. Since each state transition of the hexapod robot is checked by the proposed free gait transition feasibility model, the coxa angle is always within the angle range as shown in Table 3.

Fig. 9 shows the heat map of the passable area of the optimal free gait policy trained by the Hybrid-MASAC algorithm and the optimal tripod gait policy trained by the SAC-Tripod algorithm in the environment $E_3$ with $N_p = 150$. The heat map covers the passable area of the COM in the environment, and the darker the color, the greater the probability of passability. Fig. 9(a) shows the heat map of the passable area using free gait, and Fig. 9(b) shows the heat map of the passable area using tripod gait. It can be observed that the distribution of the passable area with free gait is rela-

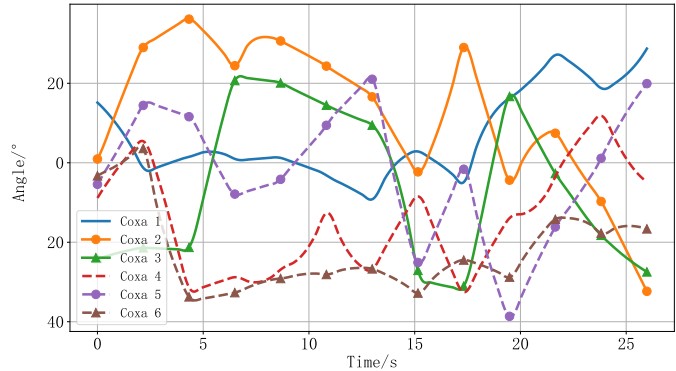

Figure 8: The coxa angles of the robot in $E_3$ with $N_p = 150$.

Table 5: The Average Success Rate (ASR) of the trained policy in $E_3$ when transferred to $E_1$, $E_2$, and another random $E_3$.

|  | Direct transfer | Fine tune for 100 episodes | Fine tune for 500 episodes |
| --- | --- | --- | --- |
| $E_1$ | $6\%_{\pm 1\%}$ | $26\%_{\pm 5\%}$ | $58\%_{\pm 4\%}$ |
| $E_2$ | $4\%_{\pm 1\%}$ | $33\%_{\pm 6\%}$ | $65\%_{\pm 3\%}$ |
| $E_3$ | $62\%_{\pm 3\%}$ | $84\%_{\pm 2\%}$ | $90\%_{\pm 2\%}$ |

tively uniform, and there are more impassable areas with fixed tripod gait, which further proves that the Hybrid-MASAC algorithm is more adaptable to the complex environment than the SAC-Tripod algorithm.

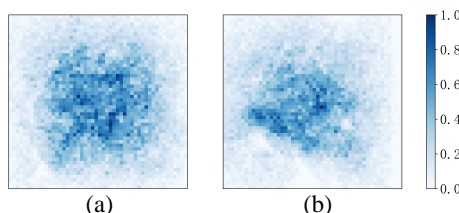

(a)         (b)

Figure 9: Passable areas of COM in (a) free gait and (b) tripod gait.

Table 5 shows the performance comparison of the algorithm trained in $E_3$ when transferred to $E_1$, $E_2$, and another random $E_3$, with performance evaluated based on whether fine-tuning is conducted. The trained policy can still achieve a success rate of around 62% in a completely new $E_3$ environment. After only 100 episodes of fine-tuning, the success rate can reach 84%, further proves the robustness of our method in the same type of environment. However, when transferred to $E_1$ and $E_2$, the accuracy of the policy significantly decreases, which may be alleviated by more advanced transfer learning algorithms. We will leave this as future work.

