# OpenReview forum: "Multi-agent Reinforcement Learning with Hybrid Action Space for Free Gait Motion Planning of Hexapod Robots"
_robot-learning.org/CoRL/2024/Conference — CoRL 2024_

### Official Review · Reviewer_XYuJ · 2024-07-19
**Review  of CoRL Submission291**

**Originality:** 3
**Technical Quality:** 3
**Clarity Of Presentation:** 2
**Potential Impact:** 3
**Recommendation:** 2
**Confidence:** 3

**Review:**

•	Please explain how to derive the equation (1)
•	You should give more explanation of each reward term in B.3. A mathematical derivation is acquired to clarify these reward terms.
•	Please indicate how to define conditions of truncated or terminated of an episode.
•	In C.2, you utilised polyhedron to simplify the kinematics constraint of legs. However, you should clarify the method to convert point cloud to polyhedron so that polyhedron is bounded inside the point cloud. Regarding Fig. 6, it seems to show that the polyhedron is expanded out of the point cloud.
•	In 3.1, line 140, you referred Transition Feasibility Model to assess the transition feasibility between the two states. However, the paper is lacking in providing how to define whether the transition is feasible or not. This is a major contribution of your paper so you should clarify this point concretely.
•	Regarding hybrid action space (3.1 line 131), it mentioned that a continuous action (foothold) governs the selection of the next foothold. However, the foothold is supposed to pick a finite number of piles nearest the robot support foot respecting to your environment condition. Thus, this action is doubted to be considered a continuous one. Please clarify this point carefully to consider whether this action is continuous or discrete.
•	You should explain why multi-agent is more efficient than single-agent to implement your system. A comparison result between these environments are suggested to enhance your proposed method.

**Quality Of The Limitations Section:**

2

**Questions For Rebuttal:**

The author should consider revising the following issues
•	Clarify how to define the Transition Feasibility Model concretely.
•	Explain and derive each term of the reward function.
•	Clarify whether the action of the foothold during the swing phase is continuous or discrete.
•	Verify the polyhedron is bounded inside the point cloud.

**Robotics Focus:**

4

**Summary Of Paper:**

The paper introduces a novel free gait motion planning method for hexapod robots, treating each leg as an independent agent with the collective goal of moving the robot's center of mass (COM) through an uneven plum blossom pile environment. The method models leg movements as a Markov game, defined by a free gait Time-Free Markov (TFM) model, and utilizes the Hybrid Multi-Agent Soft Actor-Critic (Hybrid-MASAC) algorithm to train and generate optimal hybrid policies for each leg. These policies enable the creation of optimal gait patterns, COM trajectories, and foothold sequences while meeting all kinematic and dynamic constraints. The feasibility and efficiency of the approach are validated through experiments in both simulated and real-world settings. Future research aims to enhance policy robustness and integrate the method with localization and map-building techniques.

**Summary Of Recommendation:**

The paper proposes a novel free gait motion planning method for hexapod robots, treating each leg as an independent agent with the collective goal of moving the robot's center of mass (COM) through an uneven plum blossom pile environment. Despite the innovative approach, the paper lacks clarity on several significant points that are crucial to understanding the contributions of the proposed method. To enhance the paper's contribution and impact, the author should address and revise these issues comprehensively.

---

### Official Review · Reviewer_z2Hc · 2024-07-21
**Review of paper 291**

**Originality:** 3
**Technical Quality:** 3
**Clarity Of Presentation:** 4
**Potential Impact:** 3
**Recommendation:** 3
**Confidence:** 4

**Review:**

The method is presented in sufficient detail and is written well so it is easy to understand. The problem formulation and experimental set up are described well. The results presented clearly show the effectiveness of the proposed approach. Both simulation and real world results make it an interesting contribution.

The appendix section is also very informative with many important details.

My first concern is that the supplementary material is presented as a link which I am unable to access. Perhaps its just my connection, but why not just attach it as supplementary material? Not being able to watch the real world experiments brings forward many questions of how the sim-2-real was achieved, the paper does not give any details of methods such as domain randomization, etc.. which is often required to achieve the transfer. Was it closed loop or just open loop execution of the policy?

The mathematical notations are repeated, which can sometimes make it confusing to follow. For example \theta and \phi are both network parameters as well as part of the robot observations.

In line 257-258, one of the baselines SAC-tripod, the authors mention that they actually use PPO instead of SAC, why not just call it PPO-tripod then? Were there any concerns of trading an on-policy algorithm (PPO) for an off-policy algorithm like SAC for just this one baseline?
Was the architecture kept the same? With the attention mechanism in the Critic?

**Quality Of The Limitations Section:**

3

**Questions For Rebuttal:**

Questions
1) The approach is compared to three baselines, but none of them include a case where its a single policy with hybrid action space. Is this too challenging to train ? Any rationale for that I am missing?
2) For different environments in Table 1. The average success rate for E3 is higher than E1, although intuitively E3 is a more challenging environment. The reason stated for that is better policy exploration leading to more robust behavior, did the authors try deploying a policy trained in E3 in E1?
3) Im also curious to see how this approach would perform when various uncertainties, for example from sensor noise, transition probabilities, etc.. are considered.

**Robotics Focus:**

4

**Summary Of Paper:**

The paper presents a Multi-agent combined with hybrid action space reinforcement learning approach to address the challenge of navigating through a complex environment using a hexapod robot. The algorithm is able to automatically generate COM motion, gait and foothold sequences.

**Summary Of Recommendation:**

Weak accept, additional information on real world experiments could strengthen the paper

---

### Official Review · Reviewer_oUGJ · 2024-07-28

**Originality:** 2
**Technical Quality:** 3
**Clarity Of Presentation:** 2
**Potential Impact:** 2
**Recommendation:** 2
**Confidence:** 3

**Review:**

This paper considers free gait motion planning for hexapod robots in uneven plum blossom pile environments, by treating each leg of the hexapod robot as an individual agent and proposing the Hybrid action Multi-Agent Soft Actor Critic (Hybrid-MASAC) algorithm to solve it. It formulates the free gait motion planning of the hexapod robot as a Markov game and proposes the Hybrid-MASAC algorithm to solve the problem in the multi-agent framework. The algorithm is tested in simulations and experiments and compared with other reinforcement learning algorithms. While the work has some merits, this reviewer is left with the following concerns:

Major concerns:

1. For the motivation, the authors mentioned that “achieving a globally optimal solution can be challenging or even impossible” for continuous optimization methods. However, global optimal solution is also not guaranteed for reinforcement learning based methods. The motivation behind this is then unclear. Please clarify.

2. Similarly, the authors mentioned two primary challenges in planning gait, COM, and foothold sequences independently in the third paragraph of Section 1. However, I think these two challenges remain the same if planning gait, COM, and foothold sequences jointly. It is then unclear how the proposed algorithm solves these two challenges. Please clarify.

3. With the above two concerns, it is not clear to me why reinforcement learning based method is better than search based or optimization based methods. Please comment on this.

5. For experiments, the authors only compared the proposed algorithm with RL based methods but not search based or optimization based methods. I would be interested in seeing the comparisons with search based and optimization based methods such as the methods introduced in the second paragraph of Section 1, given the above concerns.

6. Another concern for experiments is it seems that the authors only changed the RL algorithm in the proposed algorithm to be the baselines. For example, the authors changed Hybrid-MASAC to MASAC or single-agent SAC. I would prefer to see the comparison with the other existing free gait motion planning methods, such as [24] or [a, b], instead of just changing the RL algorithm in the proposed method.

[a] Tsounis, Vassilios, et al. "Deepgait: Planning and control of quadrupedal gaits using deep reinforcement learning." IEEE Robotics and Automation Letters 5.2 (2020): 3699-3706.
[b] Liu, Yunhong, and Yuzhang Xu. "Free gait planning of hexapod robot based on improved DQN algorithm." IEEE International Conference on Civil Aviation Safety and Information Technology, 2020.

7. The proposed method is trained and tested on the same environment. How does it perform when tested in a different environment? Please comment or validate it.

Minor concerns:

1. In the abstract, the abbreviation “COM” is used directly that may be confusing.

2. The authors mentioned that “all components are independent of each other” before (1). However, it seems that continuous components depend on discrete components from (1), no?

3. Some symbols are not defined but directly used, which makes some parts of the part not easy to follow. For example, the symbol H in (3) is not defined but directly used.

4. It would be better to provide some details about how to derive from (3) to (4) for the redefinition of Soft Actor Critic. Otherwise, it is a bit unclear for readers who is not familiar with SAC.

5. It would be better to put limitations in the main paper not in the appendix.

**Quality Of The Limitations Section:**

2

**Questions For Rebuttal:**

Please see the above concerns.

**Robotics Focus:**

4

**Summary Of Paper:**

This paper considers free gait motion planning for hexapod robots in uneven plum blossom pile environments, by treating each leg of the hexapod robot as an individual agent and proposing the Hybrid action Multi-Agent Soft Actor Critic (Hybrid-MASAC) algorithm to solve it.

**Summary Of Recommendation:**

This paper uses multi-agent reinforcement learning to solve free gait motion planning for hexapod robots. However, the motivation of using reinforcement learning is not clear and there is a lack of comparison with other search or optimization based methods. Some parts of the paper need more clarifications.

---

### Author Rebuttal · Authors · 2024-08-09

We have uploaded all the supplementary materials to a zip file.

---

### Decision · Program_Chairs · 2024-09-05

**Decision:**

Accept

**Comment:**

This paper received 3 fairly confident reviews.  This paper considers free gait motion planning for hexapod robots in uneven plum blossom pile environments  by treating each leg of the hexapod robot as an individual agent and proposing the Hybrid action Multi-Agent Soft Actor Critic (Hybrid-MASAC) algorithm to solve it.  To address in rebuttal (in addition to reviewer comments): - Address the justification of the use or RL - Expand experimental trials; consider ablations proposed by reviewers - Address introduction of uncertainty. - Address clarity issues pointed out by reviewers.  Post-rebuttal: This paper uses multi-agent reinforcement learning to solve free gait motion planning for hexapod robots.  The authors have addressed many issues raised and have added comparative experiments with the traditional graph-based A algorithm* and the latest free-gait based HFG-DRL algorithm; they also included two sets of ablation experiments. The motivation of using reinforcement learning could be better demonstrated and we encourage to do so in the camera ready. A reviewer updated to weak accept, but didn't change the score -- so finally there are two weak accepts and a weak reject.